# Prevalence of targeted therapy-related genetic variations in NSCLC and their relationship with clinicopathological characteristics

**Fanghua Li**[1,2©]**, Peng Ye**[3©]**, Peiling Cai**[3]**, Dandan Dong**[2]**, Yihao Zhang**[3]**, Yue Yang**[2]**, Xingwang Sun**[1] *

**1** Department of Pathology, Affiliated Hospital of Southwest Medical University, Luzhou, China, **2** Department of Pathology, Sichuan Academy of Medical Sciences & Sichuan Provincial People's Hospital, Chengdu, China, **3** Department of Anatomy and Histology, College of Medicine, Chengdu University, Chengdu, China

© These authors contributed equally to this work.

* 2753441238@qq.com

**Data Availability Statement:** All relevant data are within the paper and its Supporting information files.

## Abstract

### Background

Non-small cell lung cancer (NSCLC) is the most common cancer type in China. Targeted therapies have been used to treat NSCLC for two decades, which is only suitable for a sub-group of patients with specific genetic variations. The aim of this study was to investigate the prevalence of genetic variations leading to sensitivity or resistance to targeted therapies in NSCLC, and their relationship with clinicopathological characteristics of the patients.

### Methods

Tumor samples were collected from 404 patients who were diagnosed to have NSCLC and underwent surgery, transthoracic biopsy, bronchoscopy biopsy, or pleural aspiration in Sichuan Provincial People's Hospital from January 2019 to March 2020. Commercial amplification-refractory mutation system kits were used to detect targeted therapy-related genetic variations in those tumor samples. The prevalence of genetic variations and their relationship with patient clinicopathological characteristics were analyzed using statistical software, followed by subgroup analysis.

### Results

In all, 50.7% of the NSCLC patients had sensitive genetic variations to anti-EGFR therapies, and 4.9% of those patients had co-existing resistant genetic variations. Fusions in *ALK*, *ROS1*, or *RET* were found in 7.7% of the patients, including 2 patients with co-existing *EGFR* exon 19 deletion or L858R. *EGFR* exon 19 deletion and L858R were more common in female patients and adenocarcinoma. Further subgroup analysis confirmed the observation in female patients in adenocarcinoma subgroup, and in adenocarcinoma in male patients. In addition, smokers were more likely to have squamous cell carcinoma and *KRAS*

**Funding:** This work was supported by 2018 Research Grant of The Education Department of Sichuan Province (No. 18CZ0044) to XS, and Co-Research Grant of Science & Technology Department of Sichuan Province and Luzhou Government and Luzhou Medical College (No. 14JC0084) to XS. The funders had no role in study design, data collection and analysis, decision to publish, or preparation of the manuscript.

**Competing interests:** Enter: The authors have declared that no competing interests exist.

mutation and less likely to have *EGFR* L858R, which were also confirmed after standardization of gender except *KRAS* mutations.

## Conclusion

Nearly half of the NSCLC patients were eligible for anti-EGFR treatments. In NSCLC, female gender and adenocarcinoma may indicate higher chance of *EGFR* exon 19 deletion or L858R, and smoking history may indicate squamous cell carcinoma and *EGFR* L858R.

## Introduction

Lung cancer is a leading cause of cancer-related death worldwide [1]. There were more than 2 million new lung cancer cases and 1.76 million lung cancer-related deaths globally in 2018 [2]. As reported by National Cancer Center of China, lung cancer is also the most common cancer type and cause of cancer-related death in China, with crude incidence of 572.6 and mortality of 458.7 in 1 million population [3]. The high mortality of lung cancer is partially related to the large proportion (more than 60%) of late-stage patients when they were firstly diagnosed [4]. Since surgery is no longer applicable for those late-stage lung cancer patients, non-surgical therapies are the mainstay of treatments, including chemotherapy, radiotherapy, targeted therapy, and immunotherapy [4].

Targeted therapeutics has emerged during the past two decades. Different from traditional chemotherapy and radiotherapy, targeted therapy is only suitable for a subgroup of patients with specific genetic variations in tumor. For example, gefitinib was approved to treat metastatic non-small cell lung cancer (NSCLC) carrying exon 19 deletion or L858R mutation in *epidermal growth factor receptor* (*EGFR*) [5, 6]. Crizotinib was approved to treat advanced NSCLC with rearranged *anaplastic lymphoma kinase* (*ALK*) or *ROS1* gene [7, 8], and dabrafenib which targets a rare mutation in lung cancer (*BRAF V600E*) was approved for the treatment of *BRAF*-positive advanced NSCLC [9]. Recently, the first KRAS-targeted therapy (Sotorasib) has been approved to treat locally advanced or metastatic NSCLC carrying *KRAS* G12C mutations [10]. At the same time, companion diagnostic tests for genetic variations in *EGFR*, *ALK*, *ROS1*, *BRAF*, or *KRAS* were also approved by the administrations, and are required before those targeted therapies are given. In addition, targeted treatments for other genetic variations in NSCLC are currently under development, including *RET* fusion [11], *human epidermal growth factor receptor 2* (*HER2*) insertion (NCT04382300), or *MET* exon 14 skipping mutations (NCT02897479). Besides the drug targets mentioned above, some genetic variations in NSCLC indicate resistance to targeted therapies. For example, NSCLC patients carrying mutations in *KRAS* are less responsive to gefitinib [12]. Similar genetic variations which indicate resistance to anti-EGFR therapies include *EGFR* exon 20 insertion, *NRAS* exon 12 or 13 mutations, and *PIK3CA* mutations [13–16].

Considering the associations between the status of those genetic variations and sensitivity/resistance of targeted therapies, it would be important to understand their prevalence in diseases. Previous studies have investigated the prevalence of individual genetic variation in NSCLC [17–20]. However, not many studies investigated the prevalence and concurrence of those genetic variations in the same patient cohort. In addition, although previous studies showed linkage between smoking and tumor mutation profile in NSCLC [21], few studies investigated their association with other clinicopathological characteristics, e.g. gender, alcohol consumption. We hypothesized that there could be significant association between patient

clinicopathological characteristics and genetic variations of tumor in NSCLC, which could help further optimize the usage of targeted therapies in patients with different clinicopathological background.

The aim of this study was to investigate the prevalence of genetic variations which indicate sensitivity or resistance to targeted therapies in NSCLC, as well as the relationships between those genetic variations and clinicopathological characteristics of patients. Our study results showed significant associations between genetic variations and age (*EGFR* L858R), gender (*EGFR* exon 19 deletion, L858R), smoking history (*EGFR* L858R, exon 20 insertion, and *KRAS* mutation), histological type (*EGFR* exon 19 deletion, L858R), and tumor grade (*ALK* fusion). In addition, further analysis also revealed significant associations between gender and *EGFR* L858R, *KRAS* mutation, histological type in non-smokers, and associations between smoking history and histological type in both male and female patients, or *EGFR* L858R in male patients.

## Material and methods

### Patients

Tumor samples were retrospectively collected from patients diagnosed with NSCLC who underwent surgery, transthoracic biopsy, bronchoscopy biopsy, or pleural aspiration in Sichuan Provincial People's Hospital from January 2019 to March 2020. The inclusion criteria included: 1) pathologically-proven NSCLC, e.g. lung adenocarcinoma, lung squamous cell carcinoma, or large cell carcinoma; 2) age $\geq$ 18 years; 3) with sufficient clinicopathological information. The exclusion criteria included: 1) samples with pathologically-proven small cell lung cancer; 2) sample collected in lung not being the primary lesion; 3) samples from NSCLC patients who received chemotherapy, radiotherapy, or targeted therapies; 4) samples from patients with unclear smoking history or clinical staging information. All those tumor samples were formalin-fixed and paraffin-embedded. After sectioning and H&E staining, pathological diagnosis for each sample was confirmed by two independent pathologists, and slides with tumor content < 30% were excluded from further analysis. This study complied with the Declaration of Helsinki and was approved by Institutional Review Board of Sichuan Provincial People's Hospital. Since this was a retrospective non-interventional study, informed consent was waived by the Institutional Review Board, and all the data were collected and analyzed anonymously.

### Amplification-refractory mutation system (ARMS)

Three to eight sections from each tumor sample were firstly de-paraffinized in xylene in a 1.5-ml Eppendorf tube. DNA and RNA were extracted using a commercial DNA/RNA co-extraction kit (FFPE DNA/RNA Extraction Kit, AmoyDx, China) following the manufacturer's instruction. After measurement of A260/A280, specific gene variations in tumor samples were determined using cFDA-approved commercial kit (Human Ten-Genetic-Variation Co-Detection Kit, AmoyDx, China) which target genetic variations in *EGFR*, *ALK*, *ROS1*, *RET*, *KRAS*, *NRAS*, *BRAF*, *HER2*, or *PIK3CA* on ABI7500 Real Time PCR system (ThermoFisher, USA) following the manufacturer's instructions.

### Statistical analysis

Statistical analysis was performed using IBM SPSS 24.0 software (IBM, USA). The genetic variation results and clinicopathological characteristics from each patient were summarized and the prevalence of positivity, negativity, or a specific subgroup was calculated by dividing the

number of positive/negative/subgroup cases by the total number of patients. Relationship between different genetic variations and patient clinicopathological characteristics (age, gender, smoking history, alcohol consumption, histological type, and tumor grade) were then analyzed using $\chi^2$ test and Fisher's exact test. Difference was considered statistically significant if $P < 0.05$. When statistical difference was observed among three or more subgroups (e.g. stage I, II, III, and IV), *post hoc* analysis was performed between each of the subgroups using $\chi^2$ test and Fisher's exact test. When significant association was observed between different clinicopathological characteristics, relationship between genetic variations and clinicopathological characteristics were further analyzed in subgroups.

# Results

## Patient characteristics

In all, there were 405 patients enrolled in this study, in which 1 patient was excluded due to treatment history of targeted therapies. Samples from the rest 404 treatment-naïve patients were finally included in our analysis. As shown in Table 1, the patients were almost evenly distributed in males and females in this cohort. Majority of the patients were non-smokers (68.6%), and did not consume alcohol regularly (87.4%). Adenocarcinoma is the dominant subtype of NSCLC (94.3%) in this cohort. The percentage of tumor grades were Grade 1, 30.7%; Grade 2, 13.9%; Grade 3, 17.1%; Grade 4, 38.4%. Pathological diagnosis of NSCLC was confirmed in all samples.

**Table 1. Demographics of patients and tumor characteristics.**

| Factors | Number | % (out of 404 patients) |
|---|---|---|
| **Age** | | |
| $\leq$ median (63) | 214 | 53.0 |
| > median (63) | 190 | 47.0 |
| **Gender** | | |
| Male | 209 | 51.7 |
| Female | 195 | 48.3 |
| **Smoking** | | |
| Smoker | 125 | 30.9 |
| Non-smoker | 277 | 68.6 |
| Unknown | 2 | 0.5 |
| **Alcohol consumption** | | |
| Yes | 51 | 12.6 |
| No | 353 | 87.4 |
| **Histological type** | | |
| Adenocarcinoma | 381 | 94.3 |
| Squamous cell carcinoma | 23 | 5.7 |
| **Tumor grade** | | |
| 1 | 124 | 30.7 |
| 2 | 56 | 13.9 |
| 3 | 69 | 17.1 |
| 4 | 155 | 38.4 |
| Primary (grade 1 or 2) | 180 | 44.6 |
| Advanced (grade 3 or 4) | 224 | 55.4 |

## Biomarker evaluation

As shown in Table 2, *EGFR* exon 19 deletion (21%) and L858R mutation (26.2%) were the most common genetic variations found in this patient cohort. The overall positive rate for those two genetic variations was 47.0% (190/404), with 1 patient carrying both *EGFR* exon 19 deletion and L858R mutation. Similarly, with 4 patients carrying dual mutations (G719X/S768I, G719X/S768I, L858R/L768I, and L858R/L861Q), the overall positive rate for those rare *EGFR* mutations were 3.7% (15/404). Two patients carrying T790M also had L858R mutation. In the 9 patients carrying *EGFR* exon 20 insertion, 1 patient also had exon 19 deletion.

In other gene variations tested, the overall positive rate for *ALK*, *ROS1*, or *RET* fusion were 7.7% (31/404), with two samples holding both *RET* fusion and sensitive mutations in *EGFR* (exon 19 deletion, or L858R). Three patients carried both *KRAS* mutations and *EGFR* exon 19 deletion, and one patient had both *BRAF* mutation and *EGFR* exon 19 deletion. In addition, three patients carried both *PIK3CA* mutation and *EGFR* mutations (2 patients with exon 19 deletions, and 1 patient with both L858R and T790M).

**Table 2. Prevalence of genetic variations.**

| Biomarkers | Number | % (out of 404 patients) |
|---|---|---|
| *EGFR* | | |
| exon 19 deletion | 85 | 21.0 |
| L858R | 106 | 26.2 |
| L861Q | 5 | 1.2 |
| G719X | 8 | 2.0 |
| S768I | 4 | 1.0 |
| exon 20 insertion | 9 | 2.2 |
| T790M | 2 | 0.5 |
| Fusion | 1 | 0.2 |
| dual mutations | 8 | 2.0 |
| *ALK* | | |
| fusion | 18 | 4.5 |
| *ROS1* | | |
| fusion | 8 | 2.0 |
| *RET* | | |
| fusion | 5 | 1.2 |
| *KRAS* | | |
| G12X, G13X | 36 | 8.9 |
| *NRAS* | | |
| Q61H | 1 | 0.2 |
| *BRAF* | | |
| V600X | 5 | 1.2 |
| *HER2* | | |
| exon 20 insertion | 14 | 3.5 |
| *PIK3CA* | | |
| H1047R/E545K | 4 | 1.0 |
| *MET* | | |
| exon 14 skipping | 6 | 1.5 |

EGFR, epidermal growth factor receptor; ALK, anaplastic lymphoma kinase; HER2, human epidermal growth factor receptor 2.

### Relationship between genetic variations in tumor and patient clinicopathological characteristics

As shown in Tables 3 and 4, older patients (age > median) were associated with significantly less prevalence of *EGFR* L858R mutation, compared to younger patients (age ≤ median). In addition, male patients showed significantly lower prevalence of *EGFR* exon 19 deletion and L858R mutation but significantly higher prevalence of *KRAS* mutations, compared to female patients. Smoking was associated with significantly lower prevalence of *EGFR* L858R mutation and *KRAS* mutation. In the two subtypes of NSCLC, adenocarcinoma was significantly associated with higher prevalence of *EGFR* exon 19 deletion and L858R mutation, compared to squamous cell carcinoma. Significant association was also observed between tumor grade and *ALK* fusion, and *post hoc* analysis showed that *ALK* fusion was significantly more prevalent in grade II tumors compared to grade IV tumors (S1 Table). No significant difference was found between alcohol consumption and histological type of NSCLC, or gene variations in tumor.

### Relationship between histological type, tumor grade, and demographics of patients

As shown in Table 5, male patients were significantly more likely to be smokers and alcohol consumers. Adenocarcinoma was significantly more prevalent in female patient and non-smokers, compared to male patients and smokers, respectively. Significant association was observed between tumor grade and age, gender, smoking. Furthermore, advanced tumor was significantly more prevalent in older patients, male patients, smokers, and alcohol consumers. *Post hoc* analysis showed that stage III and stage IV tumors were significantly more prevalent in older patients. Compared to stage I tumors, male patients and smoking patients were significantly more likely to have stage III and stage IV tumors (S1 Table).

### Association between genetic variations, histological type, tumor grade, and gender or smoking history of patients

Since the percentage of smokers was much higher in male patients than in females, we further studied the relationship among histological subtype, tumor grade, genetic variations, and gender and smoking history of patients. As shown in Table 6, in both male and female patients, smokers are significantly more likely to have squamous cell carcinoma than adenocarcinoma. In male patients, tumors from smokers were significantly less likely to carry *EGFR* L858R mutation. Interestingly, in non-smokers, male patients were more likely to have squamous cell carcinoma than females. In addition, in non-smokers, more male patients were positive in *KRAS* mutations than female patients. However, in non-smokers, more female patients were positive in *EGFR* L858R mutation than male patients.

Similarly, since histological type was significantly associated with gender of patients, we further investigated the relationship among histological type, gender, and genetic variations. As shown in Table 7, in adenocarcinoma, female patients had significantly higher prevalence of *EGFR* exon 19 deletion and L858R compared to male patients. On the other hand, in male patients, adenocarcinoma was associated with higher prevalence of *EGFR* exon 19 deletion and L858R, compared to squamous cell carcinoma.

## Discussion

Several targeted therapies have been approved for the treatment of NSCLC harboring certain genetic variations in *EGFR*, *ALK*, *ROS1*, or *BRAF* [5–9] and it is required to determine the existence of those genetic variations in tumor before targeted therapies are given to patients. Better

**Table 3. Association between genetic variations and demographics of patient.**

| | | Age | | | Gender | | | Smoking | | | Alcohol consumption | | |
|---|---|---|---|---|---|---|---|---|---|---|---|---|---|
| | | ≤ 63 | > 63 | P | Male | Female | P | Smoker | Non-smoker | P | Yes | No | P |
| *EGFR* | | | | | | | | | | | | | |
| exon 19 deletion | Positive | 41 | 44 | 0.325 | 34 | 51 | 0.015 | 21 | 63 | 0.175 | 11 | 74 | 0.921 |
| | Negative | 173 | 146 | | 175 | 144 | | 104 | 214 | | 40 | 279 | |
| L858R | Positive | 65 | 41 | 0.045 | 35 | 71 | <0.001 | 17 | 89 | <0.001 | 9 | 97 | 0.136 |
| | Negative | 149 | 149 | | 174 | 124 | | 108 | 188 | | 42 | 256 | |
| L861Q | Positive | 2 | 3 | 0.669 | 3 | 2 | 1.000 | 2 | 3 | 0.648 | 1 | 4 | 0.493 |
| | Negative | 212 | 187 | | 206 | 193 | | 123 | 274 | | 50 | 349 | |
| exon 20 insertion | Positive | 7 | 2 | 0.182 | 3 | 6 | 0.324 | 3 | 6 | 1.000 | 2 | 7 | 0.317 |
| | Negative | 207 | 188 | | 206 | 189 | | 122 | 271 | | 49 | 346 | |
| G719X | Positive | 6 | 2 | 0.291 | 5 | 3 | 0.725 | 2 | 6 | 1.000 | 0 | 8 | 0.603 |
| | Negative | 208 | 188 | | 204 | 192 | | 123 | 271 | | 51 | 345 | |
| S768I | Positive | 1 | 3 | 0.346 | 2 | 2 | 1.000 | 1 | 3 | 1.000 | 1 | 3 | 0.418 |
| | Negative | 213 | 187 | | 207 | 193 | | 124 | 274 | | 50 | 350 | |
| T790M | Positive | 1 | 1 | 1.000 | 0 | 2 | 0.232 | 0 | 2 | 1.000 | 0 | 2 | 1.000 |
| | Negative | 213 | 189 | | 209 | 193 | | 125 | 275 | | 51 | 351 | |
| fusion | Positive | 0 | 1 | 0.470 | 0 | 1 | 0.483 | 0 | 1 | 1.000 | 0 | 1 | 1.000 |
| | Negative | 214 | 189 | | 209 | 194 | | 125 | 276 | | 51 | 352 | |
| *ALK* | | | | | | | | | | | | | |
| fusion | Positive | 9 | 9 | 0.796 | 8 | 10 | 0.527 | 4 | 14 | 0.405 | 0 | 18 | 0.146 |
| | Negative | 205 | 181 | | 201 | 185 | | 121 | 263 | | 51 | 335 | |
| *ROS1* | | | | | | | | | | | | | |
| fusion | Positive | 4 | 4 | 1.000 | 3 | 5 | 0.490 | 2 | 6 | 1.000 | 1 | 7 | 1.000 |
| | Negative | 210 | 186 | | 206 | 190 | | 123 | 271 | | 50 | 346 | |
| *RET* | | | | | | | | | | | | | |
| fusion | Positive | 4 | 1 | 0.376 | 3 | 2 | 1.000 | 2 | 3 | 0.648 | 0 | 5 | 1.000 |
| | Negative | 210 | 189 | | 206 | 193 | | 123 | 274 | | 51 | 348 | |
| *KRAS* | | | | | | | | | | | | | |
| mutation | Positive | 21 | 15 | 0.499 | 28 | 8 | 0.001 | 17 | 19 | 0.028 | 7 | 29 | 0.193 |
| | Negative | 193 | 175 | | 181 | 187 | | 108 | 258 | | 44 | 324 | |
| *NRAS* | | | | | | | | | | | | | |
| Q61H | Positive | 0 | 1 | 0.470 | 1 | 0 | 1.000 | 1 | 0 | 0.311 | 1 | 0 | 0.126 |
| | Negative | 214 | 189 | | 208 | 195 | | 124 | 277 | | 50 | 353 | |
| *BRAF* | | | | | | | | | | | | | |
| V600X | Positive | 3 | 2 | 1.000 | 2 | 3 | 0.676 | 1 | 4 | 1.000 | 0 | 5 | 1.000 |
| | Negative | 211 | 188 | | 207 | 192 | | 124 | 273 | | 51 | 348 | |
| *HER2* | | | | | | | | | | | | | |
| exon 20 insertion | Positive | 7 | 7 | 0.821 | 8 | 6 | 0.680 | 3 | 11 | 0.563 | 2 | 12 | 0.693 |
| | Negative | 207 | 183 | | 201 | 189 | | 122 | 266 | | 49 | 341 | |
| *PIK3CA* | | | | | | | | | | | | | |
| H1047R/E545K | Positive | 1 | 3 | 0.346 | 2 | 2 | 1.000 | 2 | 2 | 0.591 | 2 | 2 | 0.079 |
| | Negative | 213 | 187 | | 207 | 193 | | 123 | 275 | | 49 | 351 | |
| *MET* | | | | | | | | | | | | | |
| exon 14 skipping | Positive | 3 | 3 | 1.000 | 5 | 1 | 0.217 | 4 | 2 | 0.079 | 1 | 5 | 0.558 |
| | Negative | 210 | 187 | | 203 | 194 | | 121 | 274 | | 50 | 347 | |

EGFR, epidermal growth factor receptor; ALK, anaplastic lymphoma kinase; HER2, human epidermal growth factor receptor 2.

**Table 4. Association between genetic variations and tumor characteristics.**

| | | Histological type | | | Tumor grade | | | | | | | |
|---|---|---|---|---|---|---|---|---|---|---|---|---|
| | | Adenocarcinoma | Squamous cell carcinoma | *P* | I | II | III | IV | *P* | Primary | Advanced | *P* |
| *EGFR* | | | | | | | | | | | | |
| exon 19 deletion | Positive | 85 | 0 | 0.007 | 30 | 6 | 17 | 32 | 0.180 | 36 | 49 | 0.646 |
| | Negative | 296 | 23 | | 94 | 50 | 52 | 123 | | 144 | 175 | |
| L858R | Positive | 106 | 0 | 0.003 | 34 | 17 | 20 | 35 | 0.589 | 51 | 55 | 0.391 |
| | Negative | 275 | 23 | | 90 | 39 | 49 | 120 | | 129 | 169 | |
| L861Q | Positive | 5 | 0 | 1.000 | 3 | 0 | 0 | 2 | 0.394 | 3 | 2 | 0.660 |
| | Negative | 376 | 23 | | 121 | 56 | 69 | 153 | | 177 | 222 | |
| exon 20 insertion | Positive | 9 | 0 | 1.000 | 3 | 1 | 3 | 2 | 0.548 | 4 | 5 | 1.000 |
| | Negative | 372 | 23 | | 121 | 55 | 66 | 153 | | 176 | 219 | |
| G719X | Positive | 8 | 0 | 1.000 | 3 | 0 | 2 | 3 | 0.669 | 3 | 5 | 0.737 |
| | Negative | 373 | 23 | | 121 | 56 | 67 | 152 | | 177 | 219 | |
| S768I | Positive | 4 | 0 | 1.000 | 2 | 0 | 0 | 2 | 0.597 | 2 | 2 | 1.000 |
| | Negative | 377 | 23 | | 122 | 56 | 69 | 153 | | 178 | 222 | |
| T790M | Positive | 2 | 0 | 1.000 | 0 | 0 | 0 | 2 | 0.358 | 0 | 2 | 0.505 |
| | Negative | 379 | 23 | | 124 | 56 | 69 | 153 | | 180 | 222 | |
| fusion | Positive | 1 | 0 | 1.000 | 0 | 0 | 0 | 1 | 0.657 | 0 | 1 | 1.000 |
| | Negative | 380 | 23 | | 124 | 56 | 69 | 154 | | 180 | 223 | |
| *ALK* | | | | | | | | | | | | |
| fusion | Positive | 18 | 0 | 0.613 | 4 | 6 | 5 | 3 | 0.025 | 10 | 8 | 0.337 |
| | Negative | 363 | 23 | | 120 | 50 | 64 | 152 | | 170 | 216 | |
| *ROS1* | | | | | | | | | | | | |
| fusion | Positive | 8 | 0 | 1.000 | 1 | 2 | 1 | 4 | 0.573 | 3 | 5 | 0.737 |
| | Negative | 373 | 23 | | 123 | 54 | 68 | 151 | | 177 | 219 | |
| *RET* | | | | | | | | | | | | |
| fusion | Positive | 5 | 0 | 1.000 | 0 | 1 | 1 | 3 | 0.506 | 1 | 4 | 0.387 |
| | Negative | 376 | 23 | | 124 | 55 | 68 | 152 | | 179 | 220 | |
| *KRAS* | | | | | | | | | | | | |
| mutation | Positive | 35 | 1 | 0.709 | 12 | 5 | 5 | 14 | 0.955 | 17 | 19 | 0.736 |
| | Negative | 346 | 22 | | 112 | 51 | 64 | 141 | | 163 | 205 | |
| *NRAS* | | | | | | | | | | | | |
| Q61H | Positive | 1 | 0 | 1.000 | 0 | 0 | 0 | 1 | 0.657 | 0 | 1 | 1.000 |
| | Negative | 380 | 23 | | 124 | 56 | 69 | 154 | | 180 | 223 | |
| *BRAF* | | | | | | | | | | | | |
| V600X | Positive | 5 | 0 | 1.000 | 0 | 1 | 1 | 3 | 0.506 | 1 | 4 | 0.387 |
| | Negative | 376 | 23 | | 124 | 55 | 68 | 152 | | 179 | 220 | |
| *HER2* | | | | | | | | | | | | |
| exon 20 insertion | Positive | 14 | 0 | 1.000 | 7 | 2 | 1 | 4 | 0.397 | 9 | 5 | 0.131 |
| | Negative | 367 | 23 | | 117 | 54 | 68 | 151 | | 171 | 219 | |
| *PIK3CA* | | | | | | | | | | | | |
| H1047R/E545K | Positive | 3 | 1 | 0.210 | 1 | 0 | 1 | 2 | 0.827 | 1 | 3 | 0.632 |
| | Negative | 378 | 22 | | 123 | 56 | 68 | 153 | | 179 | 221 | |
| *MET* | | | | | | | | | | | | |
| exon 14 skipping | Positive | 6 | 0 | 1.000 | 1 | 1 | 1 | 3 | 0.885 | 2 | 4 | 0.696 |
| | Negative | 374 | 23 | | 123 | 55 | 68 | 151 | | 178 | 219 | |

EGFR, epidermal growth factor receptor; ALK, anaplastic lymphoma kinase; HER2, human epidermal growth factor receptor 2.

**Table 5. Association between demographics of patients and tumor characteristics.**

| | Age | | | Gender | | | Smoking | | | Alcohol consumption | | |
|---|---|---|---|---|---|---|---|---|---|---|---|---|
| | ≤ 63 | > 63 | P | Male | Female | P | Smoker | Non-smoker | P | Yes | No | P |
| **Gender** | | | | | | | | | | | | |
| Male | 101 | 108 | 0.053 | - | - | - | 117 | 90 | <0.001 | 48 | 161 | <0.001 |
| Female | 113 | 82 | | - | - | | 8 | 187 | | 3 | 192 | |
| **Histological type** | | | | | | | | | | | | |
| Adenocarcinoma | 205 | 176 | 0.171 | 189 | 192 | <0.001 | 107 | 272 | <0.001 | 47 | 334 | 0.513 |
| Squamous cell carcinoma | 9 | 14 | | 20 | 3 | | 18 | 5 | | 4 | 19 | |
| **Tumor grade** | | | | | | | | | | | | |
| I | 82 | 42 | <0.001 | 51 | 73 | 0.019 | 24 | 100 | 0.007 | 9 | 115 | 0.159 |
| II | 34 | 22 | | 28 | 28 | | 18 | 37 | | 7 | 49 | |
| III | 27 | 42 | | 37 | 32 | | 24 | 44 | | 10 | 59 | |
| IV | 71 | 84 | | 93 | 62 | | 59 | 96 | | 25 | 130 | |
| Primary (grade 1 or 2) | 116 | 64 | <0.001 | 79 | 101 | 0.005 | 42 | 137 | 0.003 | 16 | 164 | 0.043 |
| Advanced (grade 3 or 4) | 98 | 126 | | 130 | 94 | | 83 | 140 | | 35 | 189 | |

understanding of the prevalence of those drug targets in NSCLC and their relationships may provide more guidance to clinical practices.

Our study investigated common genetic variations in *EGFR*, fusions in *ALK* or *ROS1*, and *BRAF* V600X mutations in a cohort of NSCLC patients. For *EGFR*, there are overall 50.7% of NSCLC patients harboring sensitive genetic variations to anti-EGFR therapies (including exon 19 deletion, L858R, L861Q, G719X, and S768I). This is higher than the percentage (32.3%) reported by a previous systemic review by Zhang et al. [19], which could be due to the different ethnicity of the patient cohorts in the two studies (Chinese patients *versus* mixed ethnicity of patients), or different sensitivity of measurement methods used in the two studies (ARMS

**Table 6. Comparison of *EGFR* L858R, *EGFR* exon 20 insertion, *KRAS* mutation, histological type, and tumor grade between smoker and non-smoker in different gender, and between male and female patients in smokers or non-smokers.**

| | Male | | | Female | | | Smoker | | | Non-smoker | | |
|---|---|---|---|---|---|---|---|---|---|---|---|---|
| | Smoker | Non-smoker | P | Smoker | Non-smoker | P | Male | Female | P | Male | Female | P |
| ***EGFR* L858R** | | | | | | | | | | | | |
| Positive | 14 | 21 | 0.031 | 3 | 68 | 1.000 | 14 | 3 | 0.076 | 21 | 68 | 0.039 |
| Negative | 103 | 69 | | 5 | 119 | | 103 | 5 | | 69 | 119 | |
| ***EGFR* 20ins** | | | | | | | | | | | | |
| Positive | 2 | 1 | 1.000 | 1 | 5 | 0.225 | 2 | 1 | 0.181 | 1 | 5 | 0.667 |
| Negative | 115 | 89 | | 7 | 182 | | 115 | 7 | | 89 | 182 | |
| ***KRAS* mutation** | | | | | | | | | | | | |
| Positive | 16 | 12 | 0.943 | 1 | 7 | 0.289 | 16 | 1 | 1.000 | 12 | 7 | 0.003 |
| Negative | 101 | 78 | | 7 | 180 | | 101 | 7 | | 78 | 180 | |
| **Histological type** | | | | | | | | | | | | |
| Adenocarcinoma | 101 | 86 | 0.026 | 6 | 186 | 0.004 | 101 | 6 | 0.324 | 86 | 186 | 0.040 |
| Squamous cell carcinoma | 16 | 4 | | 2 | 1 | | 16 | 2 | | 4 | 1 | |
| **Tumor stage** | | | | | | | | | | | | |
| Primary | 38 | 40 | 0.078 | 4 | 97 | 1.000 | 38 | 4 | 0.440 | 40 | 97 | 0.247 |
| Advanced | 79 | 50 | | 4 | 90 | | 79 | 4 | | 50 | 90 | |

EGFR, epidermal growth factor receptor; EGFR 20ins, EGFR exon 20 insertion.

**Table 7. Comparison of *EGFR* exon 19 deletion and L858R between adenocarcinoma and squamous cell carcinoma in different gender, and between male and female patients in adenocarcinoma or squamous cell carcinoma.**

| | *EGFR* exon 19 deletion | | | *EGFR* L858R | | |
|---|---|---|---|---|---|---|
| | Positive | Negative | *P* | Positive | Negative | *P* |
| **Male** | | | | | | |
| Adenocarcinoma | 34 | 155 | 0.050 | 35 | 154 | 0.029 |
| Squamous cell carcinoma | 0 | 20 | | 0 | 20 | |
| **Female** | | | | | | |
| Adenocarcinoma | 51 | 141 | 0.568 | 71 | 121 | 0.555 |
| Squamous cell carcinoma | 0 | 3 | | 0 | 3 | |
| **Adenocarcinoma** | | | | | | |
| Male | 34 | 155 | 0.044 | 35 | 154 | <0.001 |
| Female | 51 | 141 | | 71 | 121 | |
| **Squamous cell carcinoma** | | | | | | |
| Male | 0 | 20 | -[a] | 0 | 20 | -[a] |
| Female | 0 | 3 | | 0 | 3 | |

EGFR, epidermal growth factor receptor.

[a]Chi-square test was not applicable due to lack of positive cases of *EGFR* exon 19 deletion or *EGFR* L858R.

versus a mixture of measurement methods). In all, there were 4.9% (10/205) of those patients holding genetic variations resistant to anti-EGFR therapy, including 2 patients with *EGFR* T790M mutation, 1 patient with *EGFR* exon 20 insertion, 3 patients with *KRAS* mutations, 1 patient with *BRAF* V600X mutation (*BRAF* V600E was also indicated as resistant mechanism for Osimertinib in a previous study [22]), and 3 patients with *PIK3CA* mutations.

Prevalence of *ALK*, *ROS1*, or *RET* fusions was relatively lower (7.7%), and only 1.2% of the patients had *BRAF* mutations. *HER2* exon 20 insertion was found in 3.5% of the patients, which is higher than the prevalence (2.2%) as reported in previous study by Arcila et al. [17], possibly due to different screening strategy. Similar to the previous study, *HER2* exon 20 insertion did not co-exist with other genetic variations in *EGFR*, *ALK*, *ROS1*, *RET*, or *BRAF*, indicating a need for targeted treatments for this specific subgroup of patients. Different from the previous study by Arcila et al. [17] which find two cases with concurrent *PIK3CA* mutations and *HER2* exon insertion, we did not find any co-existing *PIK3CA* mutations in our patient cohort. *MET* exon 14 skipping mutation was found in 1.5% (6/404) of the patients in our cohort, which is similar to previous reports (1.6 ~ 2.6% of NSCLC) [18]. In our cohort, *MET* exon 14 skipping mutation did not co-exist with other genetic variations. Similarly, previous studies also showed no co-existing *MET* exon 14 skipping mutation with other driver gene variations, except amplifications of *EGFR* in 4 cases [18].

In our results, *EGFR* L858R mutation was more commonly seen in younger patients, which is conflicting with previous reports by Zhang et al. [20], possibly due to different cut-off value for younger and older patients (age of 50 in Zhang et al. *versus* median age used for cut-off value in our study). *EGFR* exon 19 deletion and L858R mutation were more commonly observed in female patients and adenocarcinoma. Similar observation has been reported between L858R mutation and gender, smoking, and histological type in a Japanese cohort [23] and a Polish cohort [24]. Further subgroup analysis confirmed the increased prevalence of those genetic variations in female patients in adenocarcinoma subgroup, and also in adenocarcinoma in male patients, which was not performed in previous studies. Similar to previous findings [23, 25, 26], smokers were more commonly found to have squamous cell carcinoma

and *KRAS* mutation, and less likely to have *EGFR* L858R mutation in our cohort, and those associations between smoking and histological type, *EGFR* L858R mutation (but not *KRAS* mutation) were also observed after standardization of gender. Similarly, squamous cell carcinoma, *KRAS* mutations, and *EGFR* L858R mutation were more commonly found in males than females. Interestingly, after further standardization of smoking status, we found that in non-smokers, male patients were more likely to have squamous cell carcinoma and *KRAS* mutations and less likely to have *EGFR* L858R mutation than female patients. Previous study by Xue et al. [26] and Ramlau et al. [24] found that *KRAS* mutation was more common in male patients in a Chinese cohort, and *EGFR* L858R mutation was more common in female patients in the Polish cohort, but further standardization of smoking status was not performed in the study. Study by Stapelfeld et al. [27] reported conflicting results that *KRAS* mutations were more frequently found in women than in men, possibly due to the different race of patients (Caucasian) investigated in that study.

In summary, our study investigated the prevalence of targeted therapy-related genetic variations in NSCLC, as well as the relationship between those genetic variations and clinicopathological characteristics of patients. Compared to previous studies, our results showed overall similar prevalence of the investigated genetic variations in NSCLC. In addition, we also observed higher prevalence of *EGFR* exon 19 deletion and L858R both in female patients and in adenocarcinoma. Different from previous studies, we further looked into subgroups and confirmed those observations in female patients in adenocarcinoma subgroup and in adenocarcinoma in male patients, which indicate that the observations above were not due to the cross-linkage between gender and histological type but independent association between gender/histological type and the genetic variations. Furthermore, we also observed that smokers were more frequently to have squamous cell carcinoma and *KRAS* mutations, and less frequently to have *EGFR* L858R. Similarly, different from previous studies, we also performed subgroup analysis and confirmed those observations (except *KRAS* mutations) after standardization of gender, further confirming the association between histological type, *EGFR* L858R mutation and smoking. Our findings further confirmed the association between gender, histological type, smoking, and sensitive genetic variations of anti-EGFR therapies, which may help clinicians and clinical investigator in the patient selection of currently-available or potential targeted therapies. Limitations of this study could be the small patient cohort which may lead to potential statistical bias. Investigations using large patient cohort are required to further confirm the findings.

## Supporting information

**S1 Table. *Post hoc* analysis results of the association between *ALK* fusion, age, gender, and smoking history of patients and tumor stage.**
(DOCX)

**S1 File. List of genetic variations and clinicopathological characteristics of patients.**
(XLSX)

## Author Contributions

**Conceptualization:** Fanghua Li.

**Data curation:** Xingwang Sun.

**Formal analysis:** Xingwang Sun.

**Funding acquisition:** Xingwang Sun.

**Investigation:** Fanghua Li.

**Methodology:** Peng Ye, Peiling Cai, Dandan Dong, Yihao Zhang, Yue Yang.

**Project administration:** Fanghua Li, Xingwang Sun.

**Validation:** Peiling Cai, Yihao Zhang, Yue Yang.

**Writing – original draft:** Fanghua Li, Peng Ye.

**Writing – review & editing:** Xingwang Sun.

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
