## [Decision Letter · Decision Letter 0]

16 Nov 2021

PONE-D-21-30951Prevalence of targeted therapy-related genetic variations in NSCLC and their relationship with clinicopathological characteristicsPLOS ONE

Dear Dr. Sun,

Thank you for submitting your manuscript to PLOS ONE. After careful consideration, we feel that it has merit but does not fully meet PLOS ONE’s publication criteria as it currently stands. Therefore, we invite you to submit a revised version of the manuscript that addresses the points raised during the review process.

We look forward to receiving your revised manuscript.

Kind regards,

Rama Krishna Kancha

Academic Editor

PLOS ONE

Additional Editor Comments (if provided):

The reviewers opined that the study is interesting but the importance limited by the smaller cohort. In addition, several limitations regarding the presentation of data and statistical analyses were raised which need to be addressed.

2. Please ensure that you have specified (1) whether consent was informed, (2) what type you obtained (for instance, written or verbal, and if verbal, how it was documented and witnessed). If your study included minors, state whether you obtained consent from parents or guardians. If the need for consent was waived by the ethics committee and (3) If you are reporting a retrospective study of medical records or archived samples, please ensure that you have discussed whether all data were fully anonymized before you accessed them and/or whether the IRB or ethics committee waived the requirement for informed consent. If patients provided informed written consent to have data from their medical records used in research, please include this information.

“This work was supported by 2018 Research Grant of The Education Department of Sichuan Province (No. 18CZ0044) to XS, and Co-Research Grant of Science & Technology Department of Sichuan Province and Luzhou Government and Luzhou Medical College (No. 14JC0084) to XS.”

We note that you have provided funding information within the Acknowledgements. Please note that funding information should not appear in the Acknowledgments section or other areas of your manuscript. We will only publish funding information present in the Funding Statement section of the online submission form.

“This work was supported by 2018 Research Grant of The Education Department of Sichuan Province (No. 18CZ0044) to XS, and Co-Research Grant of Science & Technology Department of Sichuan Province and Luzhou Government and Luzhou Medical College (No. 14JC0084) to XS.

Reviewers' comments:

Reviewer's Responses to Questions

**Comments to the Author**

1. Is the manuscript technically sound, and do the data support the conclusions?

Reviewer #1: Yes

Reviewer #2: Partly

2. Has the statistical analysis been performed appropriately and rigorously? 

Reviewer #1: Yes

Reviewer #2: I Don't Know

3. Have the authors made all data underlying the findings in their manuscript fully available?

Reviewer #1: No

Reviewer #2: Yes

4. Is the manuscript presented in an intelligible fashion and written in standard English?

Reviewer #1: Yes

Reviewer #2: No

5. Review Comments to the Author

Reviewer #1: This manuscript is by and large repeating previous NSCLC studies and is focused on clinically actionable coding aberrations and patient characteristics in this cohort of patients. The rank order of frequency for specific aberrations recapitulates previous studies. However, the total frequencies identified in this study are remarkably higher than results from other studies. The authors address this, pointing out that it may come down to methodology and differences in the size of the cohorts (this one being smaller), which is a reasonable explanation.

The interesting finding is that among non-smokers, squamous cell carcinoma patients were more often male than female. Though as pointed out by the authors, this is a small cohort and it needs to be validated.

The technology is appropriate and sound. Statistical approach is adequate.

Critique:

The study is not necessarily novel nor addressing a research question/hypothesis and it is uncertain that any new knowledge is gained from it.

The description of results does not refer to data tables. In addition, currently there are only 2 tables, and they present limited information/data. Tables are lacking data analysis results described in the results section, columns for p-values should also be included.

Supplemental data files should list all testing results so that analyses can be repeated by others.

The product number provided does not appear to be useful to find the kit online. Please provide another way to identify the kit, such as name of the AmoyDx assay.

The background should be updated to include availability of KRAS targeted therapy.

Reviewer #2: Detailed comments have been included in the attachment.

6. PLOS authors have the option to publish the peer review history of their article (what does this mean?). If published, this will include your full peer review and any attached files.

Reviewer #1: No

Reviewer #2: No

---

## [Author Response · Author response to Decision Letter 0]

24 Dec 2021

Comments from Editor:

The reviewers opined that the study is interesting but the importance limited by the smaller cohort. In addition, several limitations regarding the presentation of data and statistical analyses were raised which need to be addressed.

Response: Many thanks for your comments. We have checked up the style of our manuscript and made necessary revisions to meet the journal’s requirements. Please find the revisions in the revised manuscript.

2. Please ensure that you have specified (1) whether consent was informed, (2) what type you obtained (for instance, written or verbal, and if verbal, how it was documented and witnessed). If your study included minors, state whether you obtained consent from parents or guardians. If the need for consent was waived by the ethics committee and (3) If you are reporting a retrospective study of medical records or archived samples, please ensure that you have discussed whether all data were fully anonymized before you accessed them and/or whether the IRB or ethics committee waived the requirement for informed consent. If patients provided informed written consent to have data from their medical records used in research, please include this information.

Response: Since this is a retrospective non-interventional study, the informed consent was waived by the Institutional Review Board, and all the data were collected and analyzed anonymously. Those statements were also included in the “Patients” section of Material and Methods. Please find them in the revised manuscript.

“This work was supported by 2018 Research Grant of The Education Department of Sichuan Province (No. 18CZ0044) to XS, and Co-Research Grant of Science & Technology Department of Sichuan Province and Luzhou Government and Luzhou Medical College (No. 14JC0084) to XS.”

We note that you have provided funding information within the Acknowledgements. Please note that funding information should not appear in the Acknowledgments section or other areas of your manuscript. We will only publish funding information present in the Funding Statement section of the online submission form.

“This work was supported by 2018 Research Grant of The Education Department of Sichuan Province (No. 18CZ0044) to XS, and Co-Research Grant of Science & Technology Department of Sichuan Province and Luzhou Government and Luzhou Medical College (No. 14JC0084) to XS.

Response: Very sorry for our mistakes. We have removed the funding information from the acknowledgement section. Please find the amended funding statements as follows:

This work was supported by 2018 Research Grant of The Education Department of Sichuan Province (No. 18CZ0044) to XS, and Co-Research Grant of Science & Technology Department of Sichuan Province and Luzhou Government and Luzhou Medical College (No. 14JC0084) to XS. The funders had no role in study design, data collection and analysis, decision to publish, or preparation of the manuscript.

Response: Sorry for the missing information. We have inserted all data regarding the testing results (genetic variations results) and clinicopathological characteristics from each of the patients as supplementary information of the revised manuscript. Please find the details in S1 File.

 

Comments from Review #1:

Reviewers' comments:

Reviewer's Responses to Questions

Comments to the Author

1. Is the manuscript technically sound, and do the data support the conclusions?

Reviewer #1: Yes

Reviewer #2: Partly

2. Has the statistical analysis been performed appropriately and rigorously?

Reviewer #1: Yes

Reviewer #2: I Don't Know

3. Have the authors made all data underlying the findings in their manuscript fully available?

Reviewer #1: No

Reviewer #2: Yes

4. Is the manuscript presented in an intelligible fashion and written in standard English?

Reviewer #1: Yes

Reviewer #2: No

5. Review Comments to the Author

Reviewer #1: This manuscript is by and large repeating previous NSCLC studies and is focused on clinically actionable coding aberrations and patient characteristics in this cohort of patients. The rank order of frequency for specific aberrations recapitulates previous studies. However, the total frequencies identified in this study are remarkably higher than results from other studies. The authors address this, pointing out that it may come down to methodology and differences in the size of the cohorts (this one being smaller), which is a reasonable explanation.

The interesting finding is that among non-smokers, squamous cell carcinoma patients were more often male than female. Though as pointed out by the authors, this is a small cohort and it needs to be validated.

The technology is appropriate and sound. Statistical approach is adequate.

Response: Many thanks for your comments and kind suggestions. Please find our point-to-point response below.

Critique:

The study is not necessarily novel nor addressing a research question/hypothesis and it is uncertain that any new knowledge is gained from it.

Response: Thanks very much for your comments. We have added more words in the Introduction section to describe the rationale, knowledge gap, hypothesis, and findings of our study. Please find the revisions in the 3rd and 4th paragraphs of Introduction section of the revised manuscript.

The description of results does not refer to data tables. In addition, currently there are only 2 tables, and they present limited information/data. Tables are lacking data analysis results described in the results section, columns for p-values should also be included.

Response: Thanks very much for your suggestion. We have added five more tables (Table 3 - 7) to provide more details of the results. Relevant parts of the Results (section 3-5) and Discussion (paragraph 4) were also revised accordingly. Please find the revisions in the revised manuscript.

Supplemental data files should list all testing results so that analyses can be repeated by others.

Response: Thanks very much for your suggestion. We have listed all the testing results and clinicopathological characteristics of patients in supporting information. Please find the detailed information in S1 File.

The product number provided does not appear to be useful to find the kit online. Please provide another way to identify the kit, such as name of the AmoyDx assay.

Response: Very sorry for the misleading information provided. We have replaced the product number with the name of the AmoyDx assy kit. Please find the revisions in the 2nd section in Methods (Amplification-refractory mutation system (ARMS) section).

The background should be updated to include availability of KRAS targeted therapy.

Response: Very sorry for the missing information. Following your suggestions, we have revised the Introduction section to reflect the approval of KRAS-targeted therapy for the treatment of KRAS G12C-mutated NSCLC. Please find the revisions in the 2nd paragraph of the Introduction in the revised manuscript.

Reviewer #2: Detailed comments have been included in the attachment.

6. PLOS authors have the option to publish the peer review history of their article (what does this mean?). If published, this will include your full peer review and any attached files.

Do you want your identity to be public for this peer review? For information about this choice, including consent withdrawal, please see our Privacy Policy.

Reviewer #1: No

Reviewer #2: No

 

Comments from Review #2:

The study is aimed at assessing the prevalence of sanitizing and resistant mutations among patients with cancer in China. The authors also aimed to understand if there were any associations between genetic alterations and clinicopathological characteristics of patients with cancer. They found that nearly half of the patients (N=404) were eligible for receiving anti-EGFR therapy. Gender could influence the histological type and KRAS mutation frequency among non-smokers.

Response: Many thanks for your comments and kind suggestions. Please find our point-to-point response below.

Title: 

The title is indicative of what the study entails. 

Response: Many thanks for your comments.

Abstract: 

This should be a standalone section of the manuscript, and therefore, should adequately summarize all sections of the main text. Currently the abstract does not provide enough background or context for the study. Similarly, the methods section of the abstract could be more detailed.

Response: Many thanks for your suggestions. We have expanded the Background, Methods, Results, and Conclusion of the Abstract to provide more detailed information. Please find the revisions in the Abstract of the revised manuscript.

Introduction: 

-Although this section provides sufficient background about the incidence of lung cancer globally and in China and also describes the current literature with respect to available therapeutic targets in NSCLC, the rationale for the current study and the current knowledge gap in existing literature have not been clearly highlighted.

Response: Thanks very much for your comments and sorry for the missing information. Following your suggestions, we have added the rationale for this study and the knowledge gap in Introduction. Please find the revisions in the 3rd and 4th paragraphs in the Introduction section of the revised manuscript.

-Also, this section does not adequately bring out the originality of the current work and the need for it.

Response: Thanks very much for your comments. We have inserted the background (originality) of this study and the gap of knowledge in the Introduction section, and also discussed the need for the conduction of this study in the same section. Please find the revisions in the 3rd paragraph of the Introduction section in the revised manuscript.

Materials and methods: 

-The study design and setting have not been described in this section. The authors have also not mentioned if informed consent was obtained from the participants of this study and whether the study was performed in accordance with ethical standards laid out for studies involving human subjects.

Response: Sorry for the missing information. This was a retrospective non-interventional study and all the data were collected and analyzed anonymously. Therefore, the informed consent from the patients was waived by the Institutional Review Board. The study design complied with the Declaration of Helsinki and was approved by the Institutional Review Board. Those descriptions have been included in the “Patients” section of Materials and Methods. Please find them in the revised manuscript.

-The inclusion and exclusion criteria used for patient recruitment in the current study have not been described. 

Response: Sorry for the missing description. We have inserted the inclusion criteria and exclusion criteria in the 1st part (Patients) of Material and Methods section. Please find it in the revised manuscript.

-Since, the aim of this study was to investigate the prevalence of genetic variations which indicate either sensitivity or resistance to targeted therapies in NSCLC, it is imperative to provide details about the line of treatment received by the patients. Currently, it is not clear from this section if the patients were treatment-naive or had developed resistance to a previous line of therapy. 

Response: Sorry for the missing information. All the patients were treatment-naïve. In the exclusion criteria newly-added into the manuscript, patients who received chemotherapy, radiotherapy, or targeted therapies were excluded. We have further clarified this issue at the beginning of Results. Please find the revisions in the 1st section (Patients) of Material and Methods, and 1st section (Patient characteristics) of Results of the revised manuscript.

-This section currently does not describe the study endpoints.

Response: Sorry for the missing description. We have expanded the “Statistical analysis” section to describe the study endpoints, including prevalence of genetic variations and patient clinicopathological characteristics, and the detailed analysis on their relationships. Please find the detailed revisions in “Statistical analysis” section in Material and Methods in the revised manuscript.

Results:

This section does not provide information about how many patients were enrolled, how many were excluded, and how many were finally included in the analysis. 

Response: Sorry for the missing information. We have inserted the numbers of patients enrolled, excluded, and finally included in our study at the beginning of Results. Please find the description in the 1st section (Patient characteristics) of Results in the revised manuscript.

Discussion:

The authors have mentioned that 50.7% of NSCLC patients harbored genetic variations sensitive to anti-EGFR therapies (including exon 19 deletion, L858R, L861Q, G719X, and S768I), and that this is higher than the percentage (32.3%) reported by a previous systemic review by Zhang et al.[16]. The authors could try to provide a reason for this considerably higher proportion observed in the current study. Could this be because of the differences in the ethnicity of the two cohorts?

Response: Many thanks for your suggestion. We have further discussed the reason for the higher percentage of EGFR mutations found in our study. Please find the revisions in the 2nd paragraph of the Discussion in the revised manuscript.

The discussion section does not highlight the novelty of the current research and its possible implications on future research and clinical and/or diagnostic practices. 

Response: Many thanks for your comments and sorry for the missing information. We have summarized the novelty of our research in the last paragraph of Discussion. Please find the revisions in the revised manuscript.

The conclusions drawn from the results obtained have not been mentioned at the end of the manuscript text.

Response: Sorry for the missing description. We have further summarized our findings in the last paragraph of Discussion section. Please find the revisions in the revised manuscript.

Tables: 

The total cohort size for which the percentages have been denoted needs to be mentioned in the table headers. Moreover, there is considerable duplication of information between table 2 and “biomarker evaluation” section of the results.

Response: Sorry for the missing information. We have added the total cohort size (“out of 404 patients”) in the headers of Table 1 and 2. Please find them in the revised manuscript. In addition, we also revised the “biomarker evaluation” section and removed the duplicated information with Table 2. Please find the revisions in the revised manuscript.

---

## [Editor Report · Decision Letter 1]

6 Jan 2022

Prevalence of targeted therapy-related genetic variations in NSCLC and their relationship with clinicopathological characteristics

PONE-D-21-30951R1

Dear Dr. Sun,

We’re pleased to inform you that your manuscript has been judged scientifically suitable for publication and will be formally accepted for publication once it meets all outstanding technical requirements.

Kind regards,

Rama Krishna Kancha

Academic Editor

PLOS ONE

---

## [Editor Report · Acceptance letter]

12 Jan 2022

PONE-D-21-30951R1 

Prevalence of targeted therapy-related genetic variations in NSCLC and their relationship with clinicopathological characteristics 

Dear Dr. Sun:

I'm pleased to inform you that your manuscript has been deemed suitable for publication in PLOS ONE. Congratulations! Your manuscript is now with our production department. 

Kind regards, 

on behalf of

Dr. Rama Krishna Kancha 

Academic Editor

PLOS ONE